# ON THE WEAKNESSES OF REINFORCEMENT LEARNING FOR NEURAL MACHINE TRANSLATION

**Leshem Choshen[1], Lior Fox[2], Zohar Aizenbud[1], Omri Abend[1,3]**
[1] School of Computer Science and Engineering, [2] The Edmond and Lily Safra Center for Brain Sciences
[3] Department of Cognitive Sciences
The Hebrew University of Jerusalem
`first.last@mail.huji.ac.il, oabend@cs.huji.ac.il`

## ABSTRACT

Reinforcement learning (RL) is frequently used to increase performance in text generation tasks, including machine translation (MT), notably through the use of Minimum Risk Training (MRT) and Generative Adversarial Networks (GAN). However, little is known about what and how these methods learn in the context of MT. We prove that one of the most common RL methods for MT does not optimize the expected reward, as well as show that other methods take an infeasibly long time to converge. In fact, our results suggest that RL practices in MT are likely to improve performance only where the pre-trained parameters are already close to yielding the correct translation. Our findings further suggest that observed gains may be due to effects unrelated to the training signal, concretely, changes in the shape of the distribution curve.

## 1 INTRODUCTION

Reinforcement learning (RL) is an appealing path for advancement in Machine Translation (MT), as it allows training systems to optimize non-differentiable score functions, common in MT evaluation, as well as tackling the "exposure bias" (Ranzato et al., 2015) in standard training, namely that the model is not exposed during training to incorrectly generated tokens, and is thus unlikely to recover from generating such tokens at test time. These motivations have led to much interest in RL for text generation in general and MT in particular (see §2). Various policy gradient methods have been used, notably REINFORCE (Williams, 1992) and variants thereof (e.g., Ranzato et al., 2015; Edunov et al., 2018) and Minimum Risk Training (MRT; e.g., Och, 2003; Shen et al., 2016). Another popular use of RL is for training GANs (Yang et al., 2018; Tevet et al., 2018). Nevertheless, despite increasing interest and strong results, little is known about what accounts for these performance gains, and the training dynamics involved.

We present the following contributions. First, our theoretical analysis shows that commonly used approximation methods are theoretically ill-founded, and may converge to parameter values that do not minimize the risk, nor are local minima thereof (§2.2).

Second, using both naturalistic experiments and carefully constructed simulations, we show that performance gains observed in the literature likely stem not from making target tokens the most probable, but from unrelated effects, such as increasing the *peakiness* of the output distribution (i.e., the probability mass of the most probable tokens). We do so by comparing a setting where the reward is informative, vs. one where it is constant. In §4 we discuss this peakiness effect (PKE).

Third, we show that promoting the target token to be the mode is likely to take a prohibitively long time. The only case we find, where improvements are likely, is where the target token is among the first 2-3 most probable tokens according to the pretrained model. These findings suggest that REINFORCE (§5) and CMRT (§6) are likely to improve over the pre-trained model only under the best possible conditions, i.e., where the pre-trained model is "nearly" correct.

We conclude by discussing other RL practices in MT which should be avoided for practical and theoretical reasons, and briefly discuss alternative RL approaches that will allow RL to tackle a larger class of errors in pre-trained models (§7).

## 2   RL IN MACHINE TRANSLATION

An MT system generates tokens $y = (y_1, ..., y_n)$ from a vocabulary $V$ one token at a time. The probability of generating $y_i$ given preceding tokens $y_{<i}$ is given by $P_\theta(\cdot|x, y_{<i})$, where $x$ is the source sentence and $\theta$ are the model parameters. For each generated token $y_i$, we denote with $r(y_i; y_{<i}, x, y^{(ref)})$ the score, or reward, for generating $y_i$ given $y_{<i}$, $x$, and the reference sentence $y^{(ref)}$. For brevity, we omit parameters where they are fixed within context. For simplicity, we assume $r$ does not depend on following tokens $y_{>i}$.

We also assume there is exactly one valid target token, as de facto, training is done against a single reference (Schulz et al., 2018). In practice, either a token-level reward is approximated using Monte-Carlo methods (e.g., Yang et al., 2018), or a sentence-level (sparse) reward is given at the end of the episode (sentence). The latter is equivalent to a uniform token-level reward.

$r$ is often the negative log-likelihood, or a standard MT metric, e.g., BLEU (Papineni et al., 2002). RL's goal is to maximize the expected episode reward (denoted with $R$); i.e., to find

$$\theta^* = \arg\max_\theta \ R(\theta) = \arg\max_\theta \ \mathbb{E}_{y \sim P_\theta}[r(y)] \tag{1}$$

### 2.1   REINFORCE

For a given source sentence, and past predictions $y_{<i}$, REINFORCE (Williams, 1992) samples $k$ tokens ($k$ is a hyperparameter) $S = \left(y^{(1)}, ..., y^{(k)}\right)$ from $P_\theta$ and updates $\theta$ according to this rule:

$$\Delta\theta \propto \frac{1}{k} \sum_{i=1}^{k} r(y_i) \nabla \log(P_\theta(y_i)) \tag{2}$$

The right-hand side of equation 2 is an unbiased estimator of the gradient of the objective function, i.e., $\mathbb{E}[\Delta\theta] \propto \nabla_\theta R(\theta)$. Therefore, REINFORCE is performing a form of stochastic gradient ascent on $R$, and has similar formal guarantees. From here follows that if $R$ is constant with respect to $\theta$, then the expected $\Delta\theta$ prescribed by REINFORCE is zero. We note that $r$ may be shifted by a constant term (called a "baseline"), without affecting the optimal value for $\theta$.

REINFORCE is used in MT, text generation, and image-to-text tasks (Liu et al., 2016; Wu et al., 2018; Rennie et al., 2017; Shetty et al., 2017; Hendricks et al., 2016) – in isolation, or as a part of training (Ranzato et al., 2015). Lately, an especially prominent use for REINFORCE is adversarial training with discrete data, where another network predicts the reward (GAN). For some recent work on RL for NMT, see (Zhang et al., 2016; Li et al., 2017; Wu et al., 2017; Yu et al., 2017; Yang et al., 2018).

### 2.2   MINIMUM RISK TRAINING

The term Minimum Risk Training (MRT) is used ambiguously in MT to refer either to the application of REINFORCE to minimizing the risk (equivalently, to maximizing the expected reward, the negative loss), or more commonly to a somewhat different estimation method, which we term Contrastive MRT (CMRT) and turn now to analyzing. CMRT was proposed by Och (2003), adapted to NMT by Shen et al. (2016), and often used since (Ayana et al., 2016; Neubig, 2016; Shen et al., 2017; Edunov et al., 2018; Makarov & Clematide, 2018; Neubig et al., 2018).

The method works as follows: at each iteration, sample $k$ tokens $S = \{y_1, \ldots, y_k\}$ from $P_\theta$, and update $\theta$ according to the gradient of

$$\widetilde{R}(\theta, S) = \sum_{i=1}^{k} Q_{\theta,S}(y_i) r(y_i) = \mathbb{E}_{y \sim Q}\left[r(y)\right]$$

where

$$Q_{\theta,S}(y_i) = \frac{P(y_i)^\alpha}{\sum_{y_j \in S} P(y_j)^\alpha}$$

Commonly (but not universally), deduplication is performed, so $\widetilde{R}$ sums over a set of unique values (Sennrich et al., 2017). This changes little in our empirical results and theoretical analysis.

Despite the resemblance in definitions of $R$ (equation 1) and $\widetilde{R}$ (indeed, $\widetilde{R}$ is sometimes presented as an approximation of $R$), they differ in two important aspects. First, $Q$'s support is $S$, so increasing

$Q(y_i)$ for some $y_i$ necessarily comes at the expense of $Q(y)$ for some $y \in S$. In contrast, increasing $P(y_i)$, as in REINFORCE, may come at the expense of $P(y)$ for any $y \in V$. Second, $\alpha$ is a smoothness parameter: the closer $\alpha$ is to 0, the closer $Q$ is to be uniform.

We show in Appendix A.1 that despite its name, CMRT does not optimize $R$, nor does it optimize $\mathbb{E}[\widetilde{R}]$. That is, it may well converge to values that are not local maxima of $R$, making it theoretically ill-founded.[1] However, given CMRT popularity, the strong results it yielded and the absence of theory for explaining it, we discuss it here. Given a sample $S$, the gradient of $\widetilde{R}$ is given by

$$\nabla \widetilde{R} = \alpha \sum_{i=1}^{k} \Big( Q(y_i) \cdot r(y_i) \cdot \nabla \log P(y_i) \Big) - \mathbb{E}_Q[r] \nabla \log Z(S) \tag{3}$$

where $Z(S) = \sum_i P(y_i)^{\alpha}$. See Appendix A.2.

Comparing Equations 2 and 3, the differences between REINFORCE and CMRT are reflected again. First, $\nabla \widetilde{R}$ has an additional term, proportional to $\nabla \log Z(S)$, which yields the contrastive effect. This contrast may improve the rate of convergence since it counters the decrease of probability mass for non-sampled tokens.

Second, given $S$, the relative weighting of the gradients $\nabla \log P(y_i)$ is proportional to $r(y_i)Q(y_i)$, or equivalently to $r(y_i)P(y_i)^{\alpha}$. CMRT with deduplication sums over distinct values in $S$ (equation 3), while REINFORCE sums over all values. This means that the relative weight of the unique value $y_i$ is $\frac{r(y_i)|\{y_i \in S\}|}{k}$ in REINFORCE. For $\alpha = 1$ the expected value of these relative weights is the same, and so for $\alpha < 1$ (as is commonly used), more weight is given to improbable tokens, which could also have a positive effect on the convergence rate.[2] However, if $\alpha$ is too close to 0, $\nabla \widetilde{R}$ vanishes, as it is not affected by $\theta$. This tradeoff explains the importance of tuning $\alpha$ reported in the literature. In §6 we present simulations with CMRT, showing very similar trends as presented by REINFORCE.

## 3 MOTIVATING DISCUSSION

Implementing a stochastic gradient ascent, REINFORCE is guaranteed to converge to a stationary point of $R$ under broad conditions. However, not much is known about its convergence rate under the prevailing conditions in NMT.

We begin with a qualitative, motivating analysis of these questions. As work on language generation empirically showed, RNNs quickly learn to output very peaky distributions (Press et al., 2017). This tendency is advantageous for generating fluent sentences with high probability, but may also entail slower convergence rates when using RL to fine-tune the model, because RL methods used in text generation sample from the (pretrained) policy distribution, which means they mostly sample what the pretrained model deems to be likely. Since the pretrained model (or policy) is peaky, exploration of other potentially more rewarding tokens will be limited, hampering convergence.

Intuitively, REINFORCE increases the probabilities of successful (positively rewarding) observations, weighing updates by how rewarding they were. When sampling a handful of tokens in each context (source sentence $x$ and generated prefix $y_{<i}$), and where the number of epochs is not large, it is unlikely that more than a few unique tokens will be sampled from $P_\theta(\cdot|x, y_{<i})$. (In practice, $k$ is typically between 1 and 20, and the number of epochs between 1 and 100.) It is thus unlikely that anything but the initially most probable candidates will be observed. Consequently, REINFORCE initially raises their probabilities, even if more rewarding tokens can be found down the list.

We thus hypothesize the peakiness of the distribution, i.e., the probability mass allocated to the most probable tokens, will increase, at least in the first phase. We call this the peakiness-effect (PKE), and show it occurs both in simulations (§4.1) and in full-scale NMT experiments (§4.2).

With more iterations, the most-rewarding tokens will be eventually sampled, and gradually gain probability mass. This discussion suggests that training will be extremely sample-inefficient. We assess the rate of convergence empirically in §5, finding this to be indeed the case.

---

[1] Sakaguchi et al. (2017) discuss the relation between CMRT and REINFORCE, claiming that CMRT is a variant . Appendix A.1 shows that CMRT does not in fact optimize the same objective.

[2] Not performing deduplication (e.g. in THUMT (Zhang et al., 2017)) results in assigning higher relative weight to high-probability tokens, which may have an adverse effect on convergence rate.

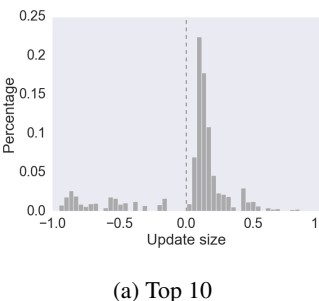
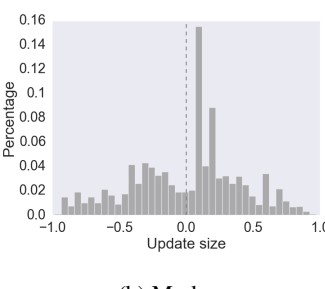

Figure 1: A histogram of the update size (x-axis) to the total predicted probability of the 10 most probable tokens (left) or the most probable token (right) in the Constant Reward setting. An update is overwhelmingly more probable to increase this probability than to decrease it.

(a) Top 10          (b) Mode

## 4 THE PEAKINESS EFFECT

We turn to demonstrate that the initially most probable tokens will initially gain probability mass, even if they are not the most rewarding, yielding a PKE.

Caccia et al. (2018) recently observed in the context of language modeling using GANs that performance gains similar to those GAN yield can be achieved by decreasing the temperature for the prediction softmax (i.e., making it peakier). However, they proposed no causes for this effect. Our findings propose an underlying mechanism leading to this trend. We return to this point in §7. Furthermore, given their findings, it is reasonable to assume that our results are relevant for RL use in other generation tasks, whose output space too is discrete, high-dimensional and concentrated.

### 4.1 CONTROLLED SIMULATIONS

We experiment with a 1-layer softmax model, that predicts a single token $i \in V$ with probability $\frac{e^{\theta_i}}{\sum_j e^{\theta_j}}$. $\theta = \{\theta_j\}_{j \in V}$ are the model's parameters. This model simulates the top of any MT decoder that ends with a softmax layer, as essentially all NMT decoders do. To make experiments realistic, we use similar parameters as those reported in the influential Transformer NMT system (Vaswani et al., 2017). Specifically, the size of $V$ (distinct BPE tokens) is 30,715, and the initial values for $\theta$ were sampled from 1,000 sets of logits taken from decoding the standard newstest2013 development set, using a pretrained Transformer model. The model was pretrained on WMT2015 training data (Bojar et al., 2015). Hyperparameters are reported in Appendix A.3. We define one of the tokens in $V$ to be the target token and denote it with $y_{best}$. We assign deterministic token reward, this makes learning easier than when relying on approximations and our predictions optimistic. We experiment with two reward functions:

1. **Simulated Reward:** $r(y) = 2$ for $y = y_{best}$, $r(y) = 1$ if $y$ is one of the 10 initially highest scoring tokens, and $r(y) = 0$ otherwise. This simulates a condition where the pretrained model is of decent but sub-optimal quality. $r$ here is at the scale of popular rewards used in MT, such as GAN-based rewards or BLEU (which are between 0 and 1).
2. **Constant Reward:** $r$ is constantly equal to 1, for all tokens. This setting is aimed to confirm that PKE is not a result of the signal carried by the reward.

Experiments with the first setting were run 100 times, each time for 50K steps, updating $\theta$ after each step. With the second setting, it is sufficient to take a single step at a time, as the expected update after each step is zero, and so any PKE seen in a single step is only accentuated in the next. It is, therefore, more telling to run more repetitions rather than more steps per initialization. We, therefore, sample 10,000 pretrained distributions, and perform a single REINFORCE step.

As RL training lasts about 30 epochs before stopping, samples about 100K tokens per epoch, and as the network already predicts $y_{best}$ in about two thirds of the contexts,[3] we estimate the number of steps used in practice to be in the order of magnitude of 1M. For visual clarity, we present figures for 50K-100K steps. However, full experiments (with 1M steps) exhibit similar trends: where REINFORCE was not close to converging after 50K steps, the same was true after 1M steps.

We evaluate the peakiness of a distribution in terms of the probability of the most probable token (the mode), the total probability of the ten most probable tokens, and the entropy of the distribution (lower entropy indicates more peakiness).

---

[3]Based on our NMT experiments, which we assume to be representative of the error rate of other systems.

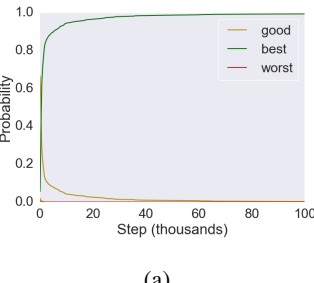 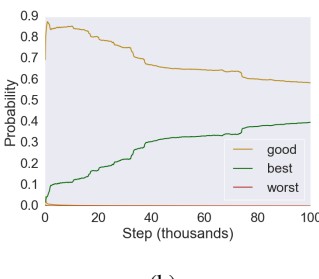 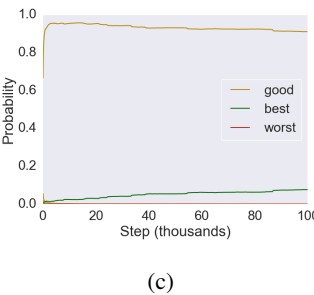

(a)          (b)          (c)

Figure 2: Token probabilities through REINFORCE training, in the controlled simulations in the Simulated Reward setting. The left/center/right figures correspond to simulations where the target token ($y_{best}$) was initially the second/third/fourth most probable token. The green line corresponds to the target token, yellow lines to medium-reward tokens and red lines to no-reward tokens.

**Results.** The distributions become peakier in terms of all three measures: on average, the mode's probability and the 10 most probable tokens increases, and the entropy decreases. Figure 1a presents the histogram of the update size, the difference in the probability of the 10 most probable tokens in the Constant Reward setting, after a single step. Figure 1b depicts similar statistics for the mode. The average entropy in the pretrained model is 2.9 is reduced to 2.85 after one REINFORCE step.

Simulated Reward setting shows similar trends. For example, entropy decreases from 3 to about 0.001 in 100K steps. This extreme decrease suggests it is effectively a deterministic policy. PKE is achieved in a few hundred steps, usually before other effects become prominent (see Figure 2), and is stronger than for Constant Reward.

## 4.2 NMT EXPERIMENTS

We turn to analyzing a real-world application of REINFORCE to NMT. Important differences between this and the previous simulations are: (1) it is rare in NMT for REINFORCE to sample from the same conditional distribution more than a handful of times, given the number of source sentences $x$ and sentence prefixes $y_{<i}$ (contexts); and (2) in NMT $P_\theta(\cdot|x, y_{<i})$ shares parameters between contexts, which means that updating $P_\theta$ for one context may influence $P_\theta$ for another.

We follow the same pretraining as in §4.1. We then follow Yang et al. (2018) in defining the reward function based on the expected BLEU score. Expected BLEU is computed by sampling suffixes for the sentence, and averaging the BLEU score of the sampled sentences against the reference.

We use early stopping with a patience of 10 epochs, where each epoch consists of 5,000 sentences sampled from the WMT2015 (Bojar et al., 2015) German-English training data. We use $k = 1$. We retuned the learning-rate, and positive baseline settings against the development set. Other hyper-parameters were an exact replication of the experiments reported in (Yang et al., 2018).

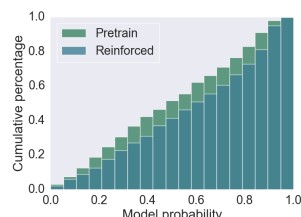

Figure 3: The cumulative distribution of the probability of the most likely token in the NMT experiments. The green distribution corresponds to the pretrained model, and the blue corresponds to the reinforced model. The y-axis is the proportion of conditional probabilities with a mode of value $\leq x$ (the x-axis). Note that a lower cumulative percentage means a more peaked output distribution. A lower cumulative percentage means a more peaked output distribution.

**Results.** Results indicate an increase in the peakiness of the conditional distributions. Our results are based on a sample of 1,000 contexts from the pretrained model, and another (independent) sample from the reinforced model.

The modes of the conditional distributions tend to increase. Figure 3 presents the distribution of the modes' probability in the reinforced conditional distributions compared with the pretrained model, showing a shift of probability mass towards higher probabilities for the mode, following RL. Another indication of the increased peakiness is the decrease in the average entropy of $P_\theta$, which was reduced from 3.45 in the pretrained model to an average of 2.82 following RL. This more modest reduction in entropy (compared to §4.1) might also suggest that the procedure did not converge to the optimal

value for $\theta$, as then we would have expected the entropy to substantially drop if not to 0 (overfit), then to the average entropy of valid next tokens (given the source and a prefix of the sentence).

## 5 PERFORMANCE FOLLOWING REINFORCE

We now turn to assessing under what conditions it is likely that REINFORCE will lead to an improvement in the performance of an NMT system. As in the previous section, we use both controlled simulations and NMT experiments.

### 5.1 CONTROLLED SIMULATIONS

We use the same model and experimental setup described in Section 4.1, this time only exploring the Simulated Reward setting, as a Constant Reward is not expected to converge to any meaningful $\theta$. Results are averaged over 100 conditional distributions sampled from the pretrained model.

Caution should be exercised when determining the learning rate (LR). Common LRs used in the NMT literature are of the scale of $10^{-4}$. However, in our simulations, no LR smaller than 0.1 yielded any improvement in $R$. We thus set the LR to be 0.1. We note that in our simulations, a higher learning rate means faster convergence as our reward is noise-free: it is always highest for the best option. In practice, increasing the learning rate may deteriorate results, as it may cause the system to overfit to the sampled instances. Indeed, when increasing the learning rate in our NMT experiments (see below) by an order of magnitude, early stopping caused the RL procedure to stop without any parameter updates.

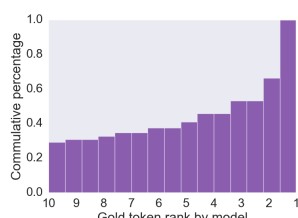

Figure 4: Cumulative percentage of contexts where the pretrained model ranks $y_{best}$ in rank $x$ or below and where it does not rank $y_{best}$ first ($x = 0$). In about half the cases it is ranked fourth or below.

Figure 2 shows the change in $P_\theta$ over the first 50K REINFORCE steps (probabilities are averaged over 100 repetitions), for a case where $y_{best}$ was initially the second, third and fourth most probable. Although these are the easiest settings, and despite the high learning rate, it fails to make $y_{best}$ the mode of the distribution within 100K steps, unless $y_{best}$ was initially the second most probable. In cases where $y_{best}$ is initially of a lower rank than four, it is hard to see any increase in its probability, even after 1M steps.

### 5.2 NMT EXPERIMENTS

We trained an NMT system, using the same procedure as in Section 4.2, and report BLEU scores over the news2014 test set. After training with an expected BLEU reward, we indeed see a minor improvement which is consistent between trials and pretrained models. While the pretrain BLEU score is 30.31, the reinforced one is 30.73.

Analyzing what words were influenced by the RL procedure, we begin by computing the cumulative probability of the target token $y_{best}$ to be ranked lower than a given rank according to the pretrained model. Results (Figure 4) show that in about half of the cases, $y_{best}$ is not among the top three choices of the pretrained model, and we thus expect it not to gain substantial probability following REINFORCE, according to our simulations.

We next turn to compare the ranks the reinforced model assigns to the target tokens, and their ranks according to the pretrained model. Figure 6 presents the difference in the probability that $y_{best}$ is ranked at a given rank following RL and the probability it is ranked there initially. Results indicate that indeed more target tokens are ranked first, and less second, but little consistent shift of probability mass occurs otherwise across the ten first ranks. It is possible that RL has managed to push $y_{best}$ in some cases between very low ranks (<1,000) to medium-low ranks (between 10 and 1,000). However, token probabilities in these ranks are so low that it is unlikely to affect the system outputs in any way. This fits well with the results of our simulations that predicted that only the initially top-ranked tokens are likely to change.

In an attempt to explain the improved BLEU score following RL with PKE, we repeat the NMT experiment this time using a constant reward of 1. Our results present a nearly identical improvement in BLEU, achieving 30.72, and a similar pattern in the change of the target tokens' ranks (see Ap-

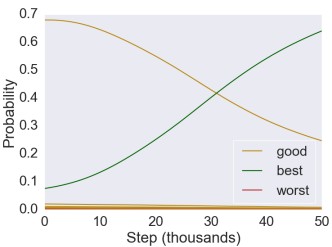 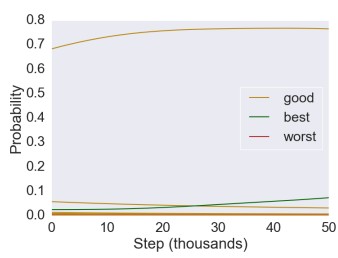

Figure 5: The probability of different tokens following CMRT, in the controlled simulations in the Simulated Reward setting. The left/right figures correspond to simulations where the target token ($y_{best}$) was initially the second/third most probable token. The green line corresponds to the target token, yellow lines to medium-reward tokens and red lines to tokens with $r(y) = 0$.

pendix 8). Therefore, there is room to suspect that even in cases where RL yields an improvement in BLEU, it may partially result from reward-independent factors, such as PKE.[4]

## 6 EXPERIMENTS WITH CONTRASTIVE MRT

In §2.2 we showed that CMRT does not, in fact, maximize $R$, and so does not enjoy the same theoretical guarantees as REINFORCE and similar policy gradient methods. However, being the RL procedure of choice in much recent work we repeat the simulations described in §4 and §5, assessing CMRT's performance in these conditions. We experiment with $\alpha = 0.005$ and $k = 20$, common settings in the literature, and average over 100 trials.

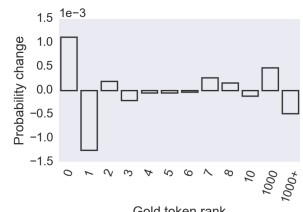

Figure 5 shows how the distribution $P_\theta$ changes over the course of 50K update steps to $\theta$, where $y_{best}$ is taken to be the second and third initially most probable token (Simulated Reward setting). Results are similar in trends to those obtained with REINFORCE: MRT succeeds in pushing $y_{best}$ to be the highest ranked token if it was initially second, but struggles where it was initially ranked third or below. We only observe a small PKE in MRT. This is probably due to the contrastive effect, which means that tokens that were not sampled do not lose probability mass.

Figure 6: Difference between the ranks of $y_{best}$ in the reinforced and the pretrained model. Each column $x$ corresponds to the difference in the probability that $y_{best}$ is ranked in rank $x$ in the reinforced model, and the same probability in the pretrained model.

All graphs we present here allow sampling the same token more than once in each batch (i.e., $S$ is a sample with replacements). Simulations with deduplication show similar results.

## 7 DISCUSSION

In this paper, we showed that the type of distributions used in NMT entail that promoting the target token to be the mode is likely to take a prohibitively long times for existing RL practices, except under the best conditions (where the pretrained model is "nearly" correct). This leads us to conclude that observed improvements from using RL for NMT are likely due either to fine-tuning the most probable tokens in the pretrained model (an effect which may be more easily achieved using reranking methods, and uses but little of the power of RL methods), or to effects unrelated to the signal carried by the reward, such as PKE. Another contribution of this paper is in showing that CMRT does not optimize the expected reward and is thus theoretically unmotivated.

A number of reasons lead us to believe that in our NMT experiments, improvements are not due to the reward function, but to artefacts such as PKE. First, reducing a constant baseline from $r$, so as to make the expected reward zero, disallows learning. This is surprising, as REINFORCE, generally and in our simulations, converges faster where the reward is centered around zero, and so the fact that this procedure here disallows learning hints that other factors are in play. As PKE can be observed even where the reward is constant (if the expected reward is positive; see §4.1), this suggests PKE

---

[4]We tried several other reward functions as well, all of which got BLEU scores of 30.73–30.84. This improvement is very stable across metrics, trials and pretrained models.

may play a role here. Second, we observe more peakiness in the reinforced model and in such cases, we expect improvements in BLEU (Caccia et al., 2018). Third, we achieve similar results with a constant reward in our NMT experiments (§5.2). Fourth, our controlled simulations show that asymptotic convergence is not reached in any but the easiest conditions (§5.1).

Our analysis further suggests that gradient clipping, sometimes used in NMT (Zhang et al., 2016; Wieting et al., 2019), is expected to hinder convergence further. It should be avoided when using REINFORCE as it violates REINFORCE's assumptions.

The per-token sampling as done in our experiments is more exploratory than beam search (Wu et al., 2018), reducing PKE. Furthermore, the latter does not sample from the behavior policy, but does not properly account for being off-policy in the parameter updates.

Adding the reference to the sample $S$, which some implementations allow (Sennrich et al., 2017) may help reduce the problems of never sampling the target tokens. However, as Edunov et al. (2018) point out, this practice may lower results, as it may destabilize training by leading the model to improve over outputs it cannot generalize over, as they are very different from anything the model assigns a high probability to, at the cost of other outputs.

# 8 CONCLUSION

The standard MT scenario poses several uncommon challenges for RL. First, the action space in MT problems is a high-dimensional discrete space (generally in the size of the vocabulary of the target language or the product thereof for sentences). This contrasts with the more common scenario studied by contemporary RL methods, which focuses mostly on much smaller discrete action spaces (e.g., video games (Mnih et al., 2015; 2016)), or continuous action spaces of relatively low dimensions (e.g., simulation of robotic control tasks (Lillicrap et al., 2015)). Second, reward for MT is naturally very sparse – almost all possible sentences are "wrong" (hence, not rewarding) in a given context. Finally, it is common in MT to use RL for tuning a pretrained model. Using a pretrained model ameliorates the last problem. But then, these pretrained models are in general quite peaky, and because training is done *on-policy* – that is, actions are being sampled from the same model being optimized – exploration is inherently limited.

Here we argued that, taken together, these challenges result in significant weaknesses for current RL practices for NMT, that may ultimately prevent them from being truly useful. At least some of these challenges have been widely studied in the RL literature, with numerous techniques developed to address them, but were not yet adopted in NLP. We turn to discuss some of them.

Off-policy methods, in which observations are sampled from a different policy than the one being currently optimized, are prominent in RL (Watkins & Dayan, 1992; Sutton & Barto, 1998), and were also studied in the context of policy gradient methods (Degris et al., 2012; Silver et al., 2014). In principle, such methods allow learning from a more "exploratory" policy. Moreover, a key motivation for using $\alpha$ in CMRT is smoothing; off-policy sampling allows smoothing while keeping convergence guarantees.

In its basic form, exploration in REINFORCE relies on stochasticity in the action-selection (in MT, this is due to sampling). More sophisticated exploration methods have been extensively studied, for example using measures for the exploratory usefulness of states or actions (Fox et al., 2018), or relying on parameter-space noise rather than action-space noise (Plappert et al., 2017).

For MT, an additional challenge is that even effective exploration (sampling diverse sets of observations), may not be enough, since the state-action space is too large to be effectively covered, with almost all sentences being not rewarding. Recently, diversity-based and multi-goal methods for RL were proposed to tackle similar challenges (Andrychowicz et al., 2017; Ghosh et al., 2018; Eysenbach et al., 2019). We believe the adoption of such methods is a promising path forward for the application of RL in NLP.

## 9 ACKNOWLEDGMENTS

This work was supported by the Israel Science Foundation (grant no. 929/17) and by the HUJI Cyber Security Research Center in conjunction with the Israel National Cyber Bureau in the Prime Minister's Office.

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

## A    APPENDIX

### A.1    CONTRASTIVE MRT DOES NOT MAXIMIZE THE EXPECTED REWARD

We hereby detail a simple example where following the Contrastive MRT method (see §2.2) does not converge to the parameter value that maximizes $R$.

Let $\theta$ be a real number in $[0, 0.5]$, and let $P_\theta$ be a family of distributions over three values $a, b, c$ such that:

$$P_\theta(x) = \begin{cases} \theta & x = a \\ 2\theta^2 & x = b \\ 1 - \theta - 2\theta^2 & x = c \end{cases}$$

Let $r(a) = 1, r(b) = 0, r(c) = 0.5$. The expected reward as a function of $\theta$ is:

$$R(\theta) = \theta + 0.5(1 - \theta - 2\theta^2)$$

$R(\theta)$ is uniquely maximized by $\theta^* = 0.25$.

Table 1 details the possible samples of size $k = 2$, their probabilities, the corresponding $\widetilde{R}$ and its gradient. Standard numerical methods show that $\mathbb{E}[\nabla \widetilde{R}]$ over possible samples $S$ is positive for $\theta \in (0, \gamma)$ and negative for $\theta \in (\gamma, 0.5]$, where $\gamma \approx 0.295$. This means that for any initialization of $\theta \in (0, 0.5]$, Contrastive MRT will converge to $\gamma$ if the learning rate is sufficiently small. For $\theta = 0$, $\widetilde{R} \equiv 0.5$, and there will be no gradient updates, so the method will converge to $\theta = 0$. Neither of these values maximizes $R(\theta)$.

We note that by using some $g(\theta)$ the $\gamma$ could be arbitrarily far from $\theta^*$. $g$ could also map to $(-inf, inf)$ more often used in neural networks parameters.

We further note that resorting to maximizing $\mathbb{E}[\widetilde{R}]$ instead, does not maximize $R(\theta)$ either. Indeed, plotting $\mathbb{E}[\widetilde{R}]$ as a function of $\theta$ for this example, yields a maximum at $\theta \approx 0.32$.

| $S$ | $P(S)$ | $\widetilde{R}$ | $\nabla \widetilde{R}$ |
|---|---|---|---|
| $\{a, b\}$ | $4\theta^3$ | $\frac{1}{1+2\theta}$ | $\frac{-2}{(1+2\theta)^2}$ |
| $\{a, c\}$ | $2\theta(1\text{-}\theta\text{-}2\theta^2)$ | $0.5 + \frac{\theta}{2-4\theta^2}$ | $\frac{2x^2+1}{2(1-2\theta^2)^2}$ |
| $\{b, c\}$ | $4\theta^2(1\text{-}\theta\text{-}2\theta^2)$ | $\frac{1-\theta-2\theta^2}{2-2\theta}$ | $\frac{\theta^2-2\theta}{(1-\theta)^2}$ |
| $a, a$ | $\theta^2$ | $1$ | $0$ |
| $b, b$ | $4\theta^4$ | $0$ | $0$ |
| $c, c$ | $(1\text{-}\theta\text{-}2\theta^2)^2$ | $0.5$ | $0$ |

Table 1: The gradients of $\widetilde{R}$ for each possible sample $S$. The batch size is $k = 2$. Rows correspond to different sampled outcomes. $\nabla \widetilde{R}$ is the gradient of $\widetilde{R}$ given the corresponding value for $S$.

### A.2    DERIVING THE GRADIENT OF $\widetilde{R}$

Given $S$, recall the definition of $\widetilde{R}$:

$$\widetilde{R}(\theta, S) = \sum_{i=1}^{k} Q_{\theta, S}(y_i) r(y_i)$$

Taking the deriviative w.r.t. $\theta$:

$$\sum_{i=1}^{k} r(y_i) \frac{\nabla P(y) \cdot \alpha P(y)^{\alpha-1} \cdot Z(S) - \nabla Z(S) \cdot P(y)^{\alpha}}{Z(S)^2} =$$

$$\sum_{i=1}^{k} r(y_i) \Big( \frac{\alpha \nabla P(y_i)}{P(y_i)} Q(y_i) - \frac{\nabla Z(S)}{Z(S)} Q(y_i) \Big) =$$

$$\sum_{i=1}^{k} r(y_i) Q(y_i) \Big( \alpha \nabla \log P(y_i) - \nabla \log Z(S) \Big) =$$

$$\alpha \sum_{i=1}^{k} \Big( r(y_i) Q(y_i) \nabla \log P(y_i) \Big) - \mathbb{E}_Q[r] \nabla \log Z(S)$$

(a)                    (b)                    (c)

Figure 7: The probability of different tokens following REINFORCE, in the controlled simulations in the Constant Reward setting. The left/center/right figures correspond to simulations where the target token ($y_{best}$) was initially the second/third/fourth most probable token. The green line corresponds to the target token, yellow lines to medium-reward tokens and red lines to tokens with $r(y) = 0$.

## A.3 NMT Implementation Details

True casing and tokenization were used (Koehn et al., 2007), including escaping html symbols and "-" that represents a compound was changed into a separate token of =. Some preprocessing used before us converted the latter to *##AT##-##AT##* but standard tokenizers in use process that into 11 different tokens, which over-represents the significance of that character when BLEU is calculated. BPE (Sennrich et al., 2016) extracted 30,715 tokens. For the MT experiments we used 6 layers in the encoder and the decoder. The size of the embeddings was 512. Gradient clipping was used with size of 5 for pre-training (see Discussion on why not to use it in training). We did not use attention dropout, but 0.1 residual dropout rate was used. In pretraining and training sentences of more than 50 tokens were discarded. Pretraining and training were considered finished when BLEU did not increase in the development set for 10 consecutive evaluations, and evaluation was done every 1,000 and 5,000 for batches of size 100 and 256 for pretraining and training respectively. Learning rate used for rmsprop (Tieleman & Hinton, 2012) was 0.01 in pretraining and for adam (Kingma & Ba, 2015) with decay was 0.005 for training. 4,000 learning rate warm up steps were used. Pretraining took about 7 days with 4 GPUs, afterwards, training took roughly the same time. Monte Carlo used 20 sentence rolls per word.

## A.4 Detailed Results for Constant Reward Setting

We present graphs for the constant reward setting in Figures 8 and 7. Trends are similar to the ones obtained for the Simulated Reward setting.

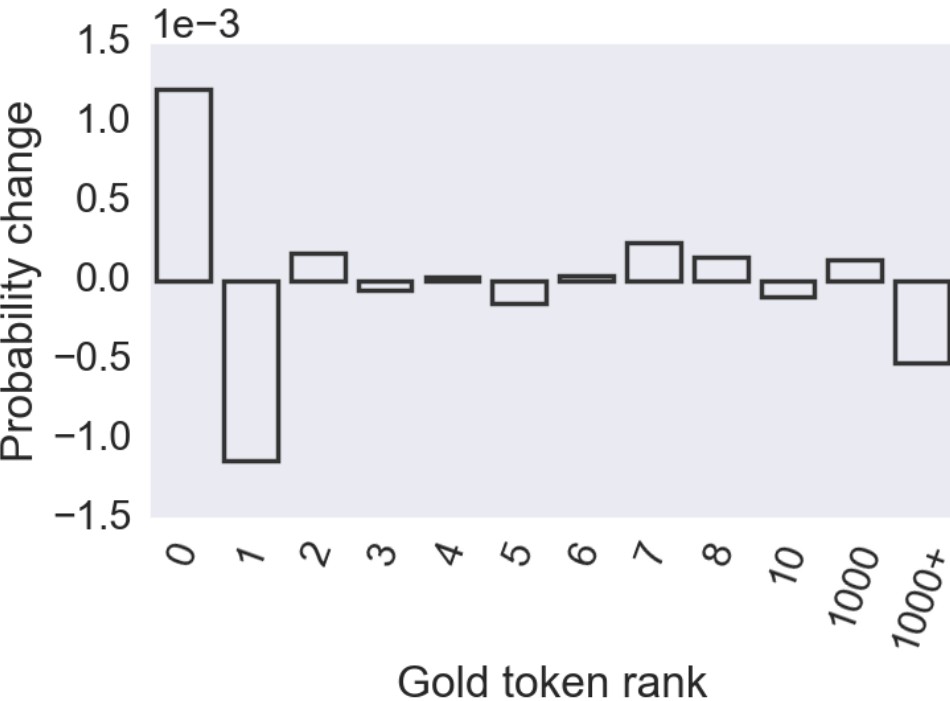

Figure 8: Difference between the ranks of $y_{best}$ in the reinforced with constant reward and the pretrained model. Each column $x$ corresponds to the difference in the probability that $y_{best}$ is ranked in rank $x$ in the reinforced model, and the same probability in the pretrained model.

