# OpenReview forum: "On the Weaknesses of Reinforcement Learning for Neural Machine Translation"
_ICLR.cc/2020/Conference — Accept (Poster)_

### Official Review · AnonReviewer3 · 2019-10-23
**Official Blind Review #3**

**Rating:** 3

**Review:**

This work carefully studies RL for neural machine translation and draws several conclusions:
1. One of the most common RL methods for MT does not optimize the expected reward, as well as show that other methods take an infeasibly long time to converge.
2. RL practices in MT are likely to improve performance only where the pre-trained parameters are already close to
yielding the correct translation.
3. Observed gains may be due to effects unrelated to the training signal, concretely, changes in the shape of the distribution curve.

I have questions about several technical claims, which lead to my doubt about the technical correctness of the paper:

	1. [updated after checking authors' response] "it may well converge to values that are not local maxima of R, making it theoretically ill-founded." Previously I had concerns about the convergence of REINFORCE with deep NNs as its policy. After checking the references provided by the authors, the local convergence can indeed be guaranteed.

	2. "reducing a constant baseline from r, so as to make the expected reward zero, disallows learning." This conflicts with my intuition. Where is the experiment supporting this claim?

	3. "MRT succeeds in pushing ybest to be the highest ranked token if it was initially second, but struggles where it was initially ranked third or below." Why ybest in third position cannot be boosted while second can be boosted? Figure 5 only shows two tokens, which cannot lead to any meaningful statistical conclusions. I'd like to see the statistic numbers:
		a. how many ybest in second are boosted to first in training? How many are not?
		b. how many ybest in third are boosted to first or second in training? How many are not?
How about other positions?

**Experience Assessment:**

I have published in this field for several years.

**Review Assessment: Checking Correctness Of Derivations And Theory:**

I assessed the sensibility of the derivations and theory.

**Review Assessment: Checking Correctness Of Experiments:**

I assessed the sensibility of the experiments.

**Review Assessment: Thoroughness In Paper Reading:**

I read the paper at least twice and used my best judgement in assessing the paper.

---

> ### Author Response · Authors · 2019-11-09
> **Response to Reviewer 3**
>
> Thank you for your comments.
>
> We were having a hard time understanding the claim behind the first question posed. Do you refer to the fact that REINFORCE only estimates a gradient? But so is SGD. The problem with MRT is that following the (stochastic) gradient of \tilde{R} may converge to a point which is not a stationary point of R, and so cannot be claimed to be maximizing R (unlike REINFORCE, under broad conditions).
>
> Or are you referring to the choice of function approximation? If so, the framing is, of course, a maximum with respect to the available parameter search space. If this is not what you were referring to, could you elaborate a bit more on this?
>
> As we have mentioned in our response to R2, and as we wrote in the paper, the model’s failing to learn with a baseline surprised us as well and is not the common case. It is not, however, unimaginable. Variance reduction results in less exploration, which we know is needed in this case. This may, or may not be the reason that baseline methods did not improve in any way over the pretrained model. We will make sure to elaborate on this point in the paper.
>
> Lastly, regarding which tokens are learnt in the simulation scenario, we can recalculate and count the numbers of changed ones. We present in figure 6 results for NMT experiments that show that the number of words placed third does not change much.

---

### Official Review · AnonReviewer2 · 2019-10-23
**Official Blind Review #2**

**Rating:** 6

**Review:**

In the context of neural machine translation, limitations of some reinforcement learning methods, in particular REINFORCE and contrastive minimum risk training (MRT), are analyzed. The authors argue that MRT doesn't optimize the expected reward. Moreover, they show that using REINFORCE, with either realistic or dummy constant rewards, may lead to a peakier distribution. Similar BLEU scores are obtained with either type of rewards, which is an interesting and perplexing result (in my opinion). For both REINFORCE and MRT, the paper shows that unless the gold token was already amongst the most probable after pre-training, it takes many samples for it to become the most likely output, which limits the usefulness of on-policy RL approaches.

I lean slightly towards rejection because the scope of the paper is somewhat too limited. The experimental section mostly covers REINFORCE without a baseline (except for validation in section 4.2, but no results are shown varying the baseline), as well as MRT in a restricted scenario. However, the analysis may still be beneficial to the community.

Establishing that minimum risk training doesn't optimize the expected reward is a valuable observation. Can the optimized and expected rewards be arbitrarily far?

The experiments demonstrating the sample inefficiency of REINFORCE (without baseline) and MRT when the best tokens have low initial probability (relative to other tokens) raise important questions about the effectiveness of these approaches.

The analysis of REINFORCE assumes non-negative rewards, which likely contributes to the peakiness effect (PKE). I would assume the peakiness effect to be mostly neutral with normalized rewards, or diminish with negative rewards (on average). It is unclear how different baselines would affect the results.

For MRT, only synthetic experiments are run. Given that the peakiness effect is much weaker than for REINFORCE, it would be useful to know results (BLEU) on the same MT task.

Other questions and remarks:

In appendix A.1, the parameter value is bounded (to maintain a valid probability distribution), which is generally not the case within a neural network. Does this distinction matter?

Do you have the RL learning curves? What could we learn from them?

It is unclear how much the claims generalize to other methods such as actor-critic (Bahdanau et al. An Actor-Critic Algorithm for Sequence Prediction).

In appendix A.1, some values are wrong. Do these affect the final result?
For example, P({a,c}) should be $2 \theta (1-\theta-2 \theta^2)$.
For S={a,c}, \nabla \tilde{R} should be $\frac{1+2 \theta^2}{2(1-2 \theta^2)^2}$.
For S={b,c} \nabla \tilde{R} should be $\frac{\theta (\theta-2)}{(1-\theta)^2}$.

-----
Update to Review 2

I appreciate that the authors corrected appendix A.1. As a nitpick, for the gradient of the reward of {a, c}, there is a x that should be replaced by \theta.

While the scope of the paper is somewhat limited, the theoretical contributions about MRT are valuable. The experiments would be more convincing with stronger baselines, but the current paper may already generate useful discussions within the community. As such, I updated my score to "Weak accept", although I won't fight for acceptance.


**Experience Assessment:**

I have published one or two papers in this area.

**Review Assessment: Checking Correctness Of Derivations And Theory:**

I assessed the sensibility of the derivations and theory.

**Review Assessment: Checking Correctness Of Experiments:**

I assessed the sensibility of the experiments.

**Review Assessment: Thoroughness In Paper Reading:**

I read the paper at least twice and used my best judgement in assessing the paper.

---

> ### Author Response · Authors · 2019-11-09
> **Response to Reviewer 2**
>
> Thank you for the kind words and for taking the time to thoroughly read and comment on our paper. A special thanks for noting the mistake in the derivation, we corrected it and it was checked and rechecked, the correction does not affect the main result and conclusion (the corrected version was uploaded).
>
> Regarding the explored settings being limited: our paper was to a large extent meant to address the current practices of applying policy gradient methods in NMT, and the explored settings were selected accordingly. There are certainly other tools in the RL toolbox to handle some of the encountered issues, and part of the claim of this paper is to indeed encourage the community to take a more sophisticated approach to this problem (as we discuss in section 8).
>
> Regarding the relation between the values that MRT and REINFORCE optimize: they can indeed be arbitrarily distant. If we take the example in Appendix A, and now write theta=g(lambda) where g is a smooth function from (-inf,inf) to (0,0.5) where g’(lambda)>0 for all values of lambda. Now assume we optimize for lambda instead. The lambda that optimizes R is just g^{-1}(theta*), but E[\nabla\tilde{R}] is positive for (-inf,g^{-1}(gamma)) and negative for (g^{-1}(gamma),inf), and so CMRT will converge to g^{-1}(gamma). For suitable choices of g, g^{-1}(gamma) and gamma(theta*) can be arbitrarily apart.
>
> The above explanation also addresses your question of whether theta being bounded in our  example makes any difference. The same argument shows that the result still holds for an unbounded parameter range.
>
> We trained NMT with a constant baseline equal to the pretrained average BLEU score over a random batch from the training set (the average does not differ too much between batches). When training with RL the score began to drop from the first epoch and when the early stopping took place, the original network without any RL training was selected as the best model. This (as you noted, and as we also noted in the paper) was surprising, and we were left to speculate why it was so. Some explanations are discussed in the paper (e.g. that the increased peakiness, coming with positive reward is a crucial part of the observed improvement, and not the reward function; see section 7, 2nd paragraph), and some are more standard conclusions (sometimes variance is a desired feature for exploration, perhaps also here). Other reasons, if exist, are left for future investigation. To conclude, other baselines (including actor-critic which may be seen as a baseline generalization) might perform better, but reducing the variance does not seem like a crucial drawback of this investigation.
>
> We don’t have the learning curves of the MT experiments, only the per-iteration scores, and these do not seem to be enlightening in any way.

---

### Official Review · AnonReviewer1 · 2019-10-25
**Official Blind Review #1**

**Rating:** 8

**Review:**

This paper first theoretically demonstrates that a commonly used reinforcement learning method for neural sequence-to-sequence models (e.g. in NMT), contrastive minimum risk training (CMRT), is not guaranteed to converge to local (let alone global) optima of the reward function. The paper then empirically demonstrates that the REINFORCE algorithm, while not subject to the same theoretical flaws as CMRT, in practice fails to improve NMT models unless the baseline model is already "nearly correct" (i.e. the correct tokens were already within the few most probable tokens before the fine-tuning steps with REINFORCE). In fact, some of the performance gains of using REINFORCE/CMRT can be attributed to making the model's output probability distribution more peaked, and not necessarily from making the target tokens more probable as commonly assumed.

Overall, this is an excellent paper that offers significant contributions for the field. I have summarised the key strengths of the paper below, along with several suggestions and questions that I hope will be addressed by the authors. Based on my assessment, I am recommending a rating of "Accept" for this paper.

Strengths:
1. The paper is very well-written and well-structured. It starts off by pointing out the theoretical limitations of CMRT (and concisely recaps the key differences between CMRT and REINFORCE), and continues with an extensive set of experiments that clearly illustrates the limitations of REINFORCE in practice.

2. I also like the use of both controlled simulations (including one where the reward is constant) and NMT experiments with real data. The controlled simulations are useful to abstract away from the full complexity of the model and investigate what happens under various control scenarios, while the NMT experiments demonstrate that the findings still hold under the realistic setup.

3. The findings are really interesting and clearly illustrate the limitations of existing REINFORCE/CMRT methods for neural sequence-to-sequence models. It is very interesting to see that REINFORCE fails to make the target token most probable when the initial model ranks the target token as the third or more probable tokens under the model (Figure 2), even under the simple controlled simulations, which highlights the prohibitively high sample complexity of the model.

4. The peakiness effect hypothesis (i.e. attributing the gains of REINFORCE to making the output distribution more peaked, and not necessarily by making the target tokens more probable) is well-supported by the paper's empirical evidence. It is really illuminating that using a constant reward of 1 leads to the same BLEU score as actually optimising for BLEU in NMT (Section 5.2).

Suggestions and questions:
1. Section 4.2 (NMT Experiments) indicates that REINFORCE fine-tuning is done for 10 epochs, with 5,000 sentences per epoch, and k=1. Considering the enormous discrete sample space, one could expect that using multi-sample REINFORCE (i.e. k > 1) and training the model for many more epochs might mitigate the identified problems to some extent, and thus change the findings. Training for 5,000 sentences * 10 epochs may just not be enough for the RL fine-tuning to make a big difference.

2. In Figure 1, the x-axis is the "Update Size" with a scale between -1.0 and 1.0. This "Update Size" variable is not really explained in the paper, and why the scale is between -1.0 and 1.0.

3. In my understanding, the controlled simulations (Section 4.1) is done at the word-level (including word-level rewards, as opposed to the NMT experiments which are done at the sequence-level with sentence-level rewards). If this is the case, this should be made clearer.

4. To make Figure 3 easier to understand, the caption should indicate that a lower cumulative percentage means a more peaked output distribution.

5. Rather than breaking down the analysis by where the target token is ranked by the initial, pre-RL model (e.g. the target token is ranked second/third best in Figures 2 and 5), perhaps what really matters is the probability assigned to the target token. For instance, even if the target token is ranked third best by the initial model, there will be a big difference whether it is assigned a probability of 0.1 or 0.01 (i.e. the latter case is much less likely to be sampled, which would exacerbate the problem). Including this analysis might help strengthen the paper further.


**Experience Assessment:**

I have read many papers in this area.

**Review Assessment: Checking Correctness Of Derivations And Theory:**

I assessed the sensibility of the derivations and theory.

**Review Assessment: Checking Correctness Of Experiments:**

I assessed the sensibility of the experiments.

**Review Assessment: Thoroughness In Paper Reading:**

I read the paper at least twice and used my best judgement in assessing the paper.

---

> ### Author Response · Authors · 2019-11-09
> **Response to Reviewer 1**
>
> Thank you for the positive review and for taking the time to thoroughly read and comment on our paper.
>
> We will, of course, address all your comments regarding changes in the last version of the paper (e.g. elaborating more on “update size”).
>
> You mentioned the number of epochs run: it is possible to run more epochs, even though it is not a short process as it is (there are Monte Carlo roll-ups involved in the process too). The experiments indeed suggest that if we want to use this method effectively, we need to run for much longer. In practice, at this point of training, the patience already stops training and results drop. Increasing the patience might work, and is worth exploring, but if it does end up helping (if no other changes are made), our results suggest it’ll take much more than a few tens of epochs.
>
> Regarding the probabilities given to the third token, it is on average 0.04. (Note that in our simulations we used those numbers so the on point 0 the frequencies mentions can be observed).

---

### Comment · AnonReviewer3 · 2019-11-15
**Only one language pair?**

Seems there is only one language pair and dataset used in experiments.
Table 6 of [1] shows that RL helps on some language pair and may hurt on other language pair. Given only one language pair in this paper, it is not clear whether or not the observations and conclusions are reliable and generalized to other languages and datasets.

[1] Google’s Neural Machine Translation System: Bridging the Gap between Human and Machine Translation

---

> ### Author Response · Authors · 2019-11-15
> **Brief response**
>
> The paper doesn't try to make a comprehensive empirical assessment of different methods for PG in NMT\text generation (that would take huge resources, varying hyperparameters, tasks, languages and more ). Instead, it hypothesizes why the application of PG in NMT as commonly practiced today is so volatile, and provides converging empirical evidence for it.  We provide several empirical results (both on simulation and real-world experiments), that lead to a similar conclusion. The simulations’ conditions are not expected to differ much between languages.

---

### Decision · Program_Chairs · 2019-12-19

**Decision:**

Accept (Poster)

**Comment:**

In my opinion, the main strength of this work is the theoretical analysis and some observations that may be of great interest to the NLP community in terms of better analyzing the performance of RL (and "RL-like") methods as optimizers. The main weakness, as pointed out by R3, the limited empirical analysis.

I would urge the authors to take R3's advice and attempt insofar as possible to broaden the scope of the empirical analysis in the final. I believe that this is important for the paper to be able to make its case convincingly.

Nonetheless, I do think that the paper makes a significant contribution that will be of interest to the community, and should be presented at ICLR. Therefore, I would recommend for it to be accepted.